# DEEP VOICE 3: SCALING TEXT-TO-SPEECH WITH CONVOLUTIONAL SEQUENCE LEARNING

**Wei Ping**[*], **Kainan Peng**[*], **Andrew Gibiansky**[*], **Sercan Ö. Arık**[*]
**Ajay Kannan**, **Sharan Narang**
Baidu Research
{pingwei01, pengkainan, gibianskyandrew, sercanarik,
 kannanajay, sharan}@baidu.com

**Jonathan Raiman**[*][†]
OpenAI
raiman@openai.com

**John Miller**[*][†]
University of California, Berkeley
miller_john@berkeley.edu

## ABSTRACT

We present Deep Voice 3, a fully-convolutional attention-based neural text-to-speech (TTS) system. Deep Voice 3 matches state-of-the-art neural speech synthesis systems in naturalness while training an order of magnitude faster. We scale Deep Voice 3 to dataset sizes unprecedented for TTS, training on more than eight hundred hours of audio from over two thousand speakers. In addition, we identify common error modes of attention-based speech synthesis networks, demonstrate how to mitigate them, and compare several different waveform synthesis methods. We also describe how to scale inference to ten million queries per day on a single GPU server.

## 1 INTRODUCTION

Text-to-speech (TTS) systems convert written language into human speech. TTS systems are used in a variety of applications, such as human-technology interfaces, accessibility for the visually-impaired, media and entertainment. Traditional TTS systems are based on complex multi-stage hand-engineered pipelines (Taylor, 2009). Typically, these systems first transform text into a compact audio representation, and then convert this representation into audio using an audio waveform synthesis method called a vocoder.

Recent work on neural TTS has demonstrated impressive results, yielding pipelines with simpler features, fewer components, and higher quality synthesized speech. There is not yet a consensus on the optimal neural network architecture for TTS. However, sequence-to-sequence models (Wang et al., 2017; Sotelo et al., 2017; Arık et al., 2017) have shown promising results.

In this paper, we propose a novel, fully-convolutional architecture for speech synthesis, scale it to very large audio data sets, and address several real-world issues that arise when attempting to deploy an attention-based TTS system. Specifically, we make the following contributions:

1. We propose a fully-convolutional character-to-spectrogram architecture, which enables fully parallel computation and trains an order of magnitude faster than analogous architectures using recurrent cells (e.g., Wang et al., 2017).

2. We show that our architecture trains quickly and scales to the LibriSpeech ASR dataset (Panayotov et al., 2015), which consists of 820 hours of audio data from 2484 speakers.

3. We demonstrate that we can generate monotonic attention behavior, avoiding error modes commonly affecting sequence-to-sequence models.

4. We compare the quality of several waveform synthesis methods, including WORLD (Morise et al., 2016), Griffin-Lim (Griffin & Lim, 1984), and WaveNet (Oord et al., 2016).

---

[*]Authors listed in reverse alphabetical order.
[†]These authors contributed to this work while members of Baidu Research.

5. We describe the implementation of an inference kernel for Deep Voice 3, which can serve up to ten million queries per day on one single-GPU server.

## 2 RELATED WORK

Our work builds upon the state-of-the-art in neural speech synthesis and attention-based sequence-to-sequence learning.

Several recent works tackle the problem of synthesizing speech with neural networks, including Deep Voice 1 (Arık et al., 2017), Deep Voice 2 (Arık et al., 2017), Tacotron (Wang et al., 2017), Char2Wav (Sotelo et al., 2017), VoiceLoop (Taigman et al., 2017), SampleRNN (Mehri et al., 2017), and WaveNet (Oord et al., 2016). Deep Voice 1 & 2 retain the traditional structure of TTS pipelines, separating grapheme-to-phoneme conversion, duration and frequency prediction, and waveform synthesis. In contrast to Deep Voice 1 & 2, Deep Voice 3 employs an attention-based sequence-to-sequence model, yielding a more compact architecture. Similar to Deep Voice 3, Tacotron and Char2Wav propose sequence-to-sequence models for neural TTS. Tacotron is a neural text-to-spectrogram conversion model, used with Griffin-Lim for spectrogram-to-waveform synthesis. Char2Wav predicts the parameters of the WORLD vocoder (Morise et al., 2016) and uses a SampleRNN conditioned upon WORLD parameters for waveform generation. In contrast to Char2Wav and Tacotron, Deep Voice 3 avoids Recurrent Neural Networks (RNNs) to speed up training. [1] Deep Voice 3 makes attention-based TTS feasible for a production TTS system with no compromise on accuracy by avoiding common attention errors. Finally, WaveNet and SampleRNN are neural vocoder models for waveform synthesis. There are also numerous alternatives for high-quality hand-engineered vocoders in the literature, such as STRAIGHT (Kawahara et al., 1999), Vocaine (Agiomyrgiannakis, 2015), and WORLD (Morise et al., 2016). Deep Voice 3 adds no novel vocoder, but has the potential to be integrated with different waveform synthesis methods with slight modifications of its architecture.

Automatic speech recognition (ASR) datasets are often much larger than traditional TTS corpora but tend to be less clean, as they typically involve multiple microphones and background noise. Although prior work has applied TTS methods to ASR datasets (Yamagishi et al., 2010), Deep Voice 3 is, to the best of our knowledge, the first TTS system to scale to thousands of speakers with a single model.

Sequence-to-sequence models (Sutskever et al., 2014; Cho et al., 2014) encode a variable-length input into hidden states, which are then processed by a decoder to produce a target sequence. An attention mechanism allows a decoder to adaptively select encoder hidden states to focus on while generating the target sequence (Bahdanau et al., 2015). Attention-based sequence-to-sequence models are widely applied in machine translation (Bahdanau et al., 2015), speech recognition (Chorowski et al., 2015), and text summarization (Rush et al., 2015). Recent improvements in attention mechanisms relevant to Deep Voice 3 include enforced-monotonic attention during training (Raffel et al., 2017), fully-attentional non-recurrent architectures (Vaswani et al., 2017), and convolutional sequence-to-sequence models (Gehring et al., 2017). Deep Voice 3 demonstrates the utility of monotonic attention during training in TTS, a new domain where monotonicity is expected. Alternatively, we show that with a simple heuristic to only enforce monotonicity during inference, a standard attention mechanism can work just as well or even better. Deep Voice 3 also builds upon the convolutional sequence-to-sequence architecture from Gehring et al. (2017) by introducing a positional encoding similar to that used in Vaswani et al. (2017), augmented with a rate adjustment to account for the mismatch between input and output domain lengths.

## 3 MODEL ARCHITECTURE

In this section, we present our fully-convolutional sequence-to-sequence architecture for TTS (see Fig. 1). Our architecture is capable of converting a variety of textual features (e.g. characters, phonemes, stresses) into a variety of vocoder parameters, e.g. mel-band spectrograms, linear-scale log magnitude spectrograms, fundamental frequency, spectral envelope, and aperiodicity parameters. These vocoder parameters can be used as inputs for audio waveform synthesis models.

---

[1]RNNs introduce sequential dependencies that limit model parallelism during training.

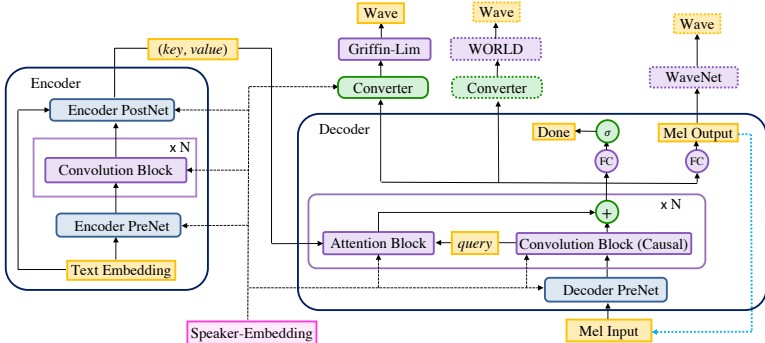

Figure 1: Deep Voice 3 uses residual convolutional layers to encode text into per-timestep *key* and *value* vectors for an attention-based decoder. The decoder uses these to predict the mel-scale log magnitude spectrograms that correspond to the output audio. (Light blue dotted arrows depict the autoregressive process during inference.) The hidden states of the decoder are then fed to a converter network to predict the vocoder parameters for waveform synthesis. See Appendix A for more details.

The Deep Voice 3 architecture consists of three components:

- **Encoder**: A fully-convolutional encoder, which converts textual features to an internal learned representation.
- **Decoder**: A fully-convolutional causal decoder, which decodes the learned representation with a multi-hop convolutional attention mechanism into a low-dimensional audio representation (mel-scale spectrograms) in an autoregressive manner.
- **Converter**: A fully-convolutional post-processing network, which predicts final vocoder parameters (depending on the vocoder choice) from the decoder hidden states. Unlike the decoder, the converter is non-causal and can thus depend on future context information.

The overall objective function to be optimized is a linear combination of the losses from the decoder (Section 3.5) and the converter (Section 3.7). We separate decoder and converter and apply multi-task training, because it makes attention learning easier in practice. To be specific, the loss for mel-spectrogram prediction guides training of the attention mechanism, because the attention is trained with the gradients from mel-spectrogram prediction besides vocoder parameter prediction.

In multi-speaker scenario, trainable speaker embeddings as in Arık et al. (2017) are used across encoder, decoder and converter. Next, we describe each of these components and the data preprocessing in detail. Model hyperparameters are available in Table 4 within Appendix C.

## 3.1 TEXT PREPROCESSING

Text preprocessing is crucial for good performance. Feeding raw text (characters with spacing and punctuation) yields acceptable performance on many utterances. However, some utterances may have mispronunciations of rare words, or may yield skipped words and repeated words. We alleviate these issues by normalizing the input text as follows:

1. We uppercase all characters in the input text.
2. We remove all intermediate punctuation marks.
3. We end every utterance with a period or question mark.
4. We replace spaces between words with special separator characters which indicate the duration of pauses inserted by the speaker between words. We use four different word separators, indicating (i) slurred-together words, (ii) standard pronunciation and space characters, (iii) a short pause between words, and (iv) a long pause between words. For example, the sentence "Either way, you should shoot very slowly," with a long pause after "way" and a short pause after "shoot", would be written as "Either way%you should shoot/very slowly%." with % representing a long pause and / representing a short pause for encoding convenience. [2]

---

[2] The pause durations can be obtained through either manual labeling or by estimated by a text-audio aligner such as Gentle (Ochshorn & Hawkins, 2017). Our single-speaker dataset is labeled by hand and our multi-speaker datasets are annotated using Gentle.

## 3.2 Joint Representation of Characters and Phonemes

Deployed TTS systems (e.g., Capes et al., 2017; Gonzalvo et al., 2016) should include a way to modify pronunciations to correct common mistakes (which typically involve proper nouns, foreign words, and domain-specific jargon). A conventional way to do this is to maintain a dictionary to map words to their phonetic representations.

Our model can directly convert characters (including punctuation and spacing) to acoustic features, and hence learns an *implicit* grapheme-to-phoneme model. This implicit conversion is difficult to correct when the model makes mistakes. Thus, in addition to character models, we also train phoneme-only models and mixed character-and-phoneme models by allowing phoneme input option explicitly. These models are identical to character-only models, except that the input layer of the encoder sometimes receives phoneme and phoneme stress embeddings instead of character embeddings.

A phoneme-only model requires a preprocessing step to convert words to their phoneme representations (by using an external phoneme dictionary or a separately trained grapheme-to-phoneme model)[3]. A mixed character-and-phoneme model requires a similar preprocessing step, except for words not in the phoneme dictionary. These out-of-vocabulary words are input as characters, allowing the model to use its implicitly learned grapheme-to-phoneme model. While training a mixed character-and-phoneme model, every word is replaced with its phoneme representation with some fixed probability at each training iteration. We find that this improves pronunciation accuracy and minimizes attention errors, especially when generalizing to utterances longer than those seen during training. More importantly, models that support phoneme representation allow correcting mispronunciations using a phoneme dictionary, a desirable feature of deployed systems.

## 3.3 Convolution Blocks for Sequential Processing

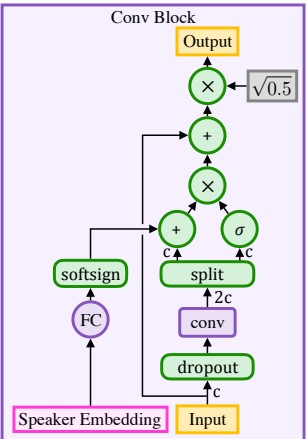

Figure 2: The convolution block consists of a 1-D convolution with a gated linear unit and a residual connection. Here $c$ denotes the dimensionality of the input. The convolution output of size $2 \cdot c$ is split into equal-sized portions: the gate vector and the input vector.

By providing a sufficiently large receptive field, stacked convolutional layers can utilize long-term context information in sequences without introducing any sequential dependency in computation. We use the convolution block depicted in Fig. 2 as the main sequential processing unit to encode hidden representations of text and audio. The convolution block consists of a 1-D convolution filter, a gated-linear unit as a learnable nonlinearity (Dauphin et al., 2017), a residual connection to the input, and a scaling factor of $\sqrt{0.5}$ [4]. The gated linear unit provides a linear path for the gradient flow, which alleviates the vanishing gradient issue for stacked convolution blocks while retaining non-linearity. To introduce speaker-dependent control, a speaker-dependent embedding is added as

---

[3]We use CMUDict 0.6b.

[4]The scaling factor ensures that we preserve the input variance early in training.

a bias to the convolution filter output, after a softsign function. We use the softsign nonlinearity because it limits the range of the output while also avoiding the saturation problem that exponential-based nonlinearities sometimes exhibit. We initialize the convolution filter weights with zero-mean and unit-variance activations throughout the entire network.

The convolutions in the architecture can be either non-causal (e.g. in encoder and converter) or causal (e.g. in decoder). To preserve the sequence length, inputs are padded with $k - 1$ timesteps of zeros on the left for causal convolutions and $(k - 1)/2$ timesteps of zeros on the left and on the right for non-causal convolutions, where $k$ is an odd convolution filter width. [5] Dropout is applied to the inputs prior to the convolution for regularization.

## 3.4 ENCODER

The encoder network (depicted in Fig. 1) begins with an embedding layer, which converts characters or phonemes into trainable vector representations, $h_e$. These embeddings $h_e$ are first projected via a fully-connected layer from the embedding dimension to a target dimensionality. Then, they are processed through a series of convolution blocks described in Section 3.3 to extract time-dependent text information. Lastly, they are projected back to the embedding dimension to create the attention *key* vectors $h_k$. The attention *value* vectors are computed from attention key vectors and text embeddings, $h_v = \sqrt{0.5}(h_k + h_e)$, to jointly consider the local information in $h_e$ and the long-term context information in $h_k$. The *key* vectors $h_k$ are used by each attention block to compute attention weights, whereas the final *context* vector is computed as a weighted average over the *value* vectors $h_v$ (see Section 3.6).

## 3.5 DECODER

The decoder (depicted in Fig. 1) generates audio in an autoregressive manner by predicting a group of $r$ future audio frames conditioned on the past audio frames. Since the decoder is autoregressive, it must use causal convolution blocks. We choose mel-band log-magnitude spectrogram as the compact low-dimensional audio frame representation. Similar to Wang et al. (2017), we empirically observed that decoding multiple frames together (i.e. having $r > 1$) yields better audio quality.

The decoder network starts with multiple fully-connected layers with rectified linear unit (ReLU) nonlinearities to preprocess input mel-spectrograms (denoted as "PreNet" in Fig. 1). Then, it is followed by a series of causal convolution and attention blocks. These convolution blocks generate the *queries* used to attend over the encoder's hidden states (see Section 3.6). Lastly, a fully-connected layer output the next group of $r$ audio frames and also a binary "final frame" prediction (indicating whether the last frame of the utterance has been synthesized). Dropout is applied before each fully-connected layer prior to the attention blocks, except for the first one. An L1 loss [6] is computed using the output mel-spectrograms and a binary cross-entropy loss is computed using the final-frame prediction.

## 3.6 ATTENTION BLOCK

We use a dot-product attention mechanism (depicted in Fig. 3) similar to Vaswani et al. (2017). The attention mechanism uses a *query* vector (the hidden states of the decoder) and the per-timestep *key* vectors from the encoder to compute attention weights, and then outputs a *context* vector computed as the weighted average of the *value* vectors.

We observe empirical benefits from introducing a inductive bias where the attention follows a monotonic progression in time. Thus, we add a positional encoding to both the key and the query vectors. These positional encodings $h_p$ are chosen as $h_p(i) = \sin\left(\omega_s i / 10000^{k/d}\right)$ (for even $i$) or $\cos\left(\omega_s i / 10000^{k/d}\right)$ (for odd $i$), where $i$ is the timestep index, $k$ is the channel index in the positional encoding, $d$ is the total number of channels in the positional encoding, and $\omega_s$ is the *position rate* of the encoding. The position rate dictates the average slope of the line in the attention distribution, roughly corresponding to speed of speech. For a single speaker, $\omega_s$ is set to one for the query, and

---

[5] We restrict to odd convolution widths to simplify the convolution arithmetic.

[6] We choose L1 loss since it yields the best result empirically. Other loss such as L2 may suffer from outlier spectral features, which may correspond to non-speech noise.

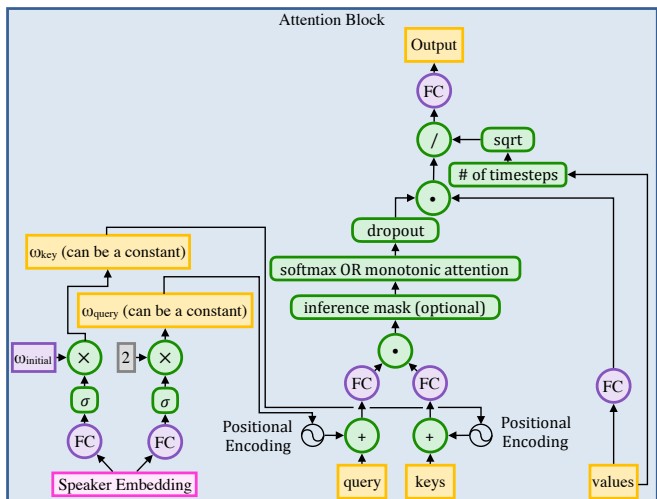

Figure 3: Positional encodings are added to both keys and query vectors, with rates of $\omega_{key}$ and $\omega_{query}$ respectively. Forced monotonicity can be applied at inference by adding a mask of large negative values to the logits. One of two possible attention schemes is used: softmax or monotonic attention from Raffel et al. (2017). During training, attention weights are dropped out.

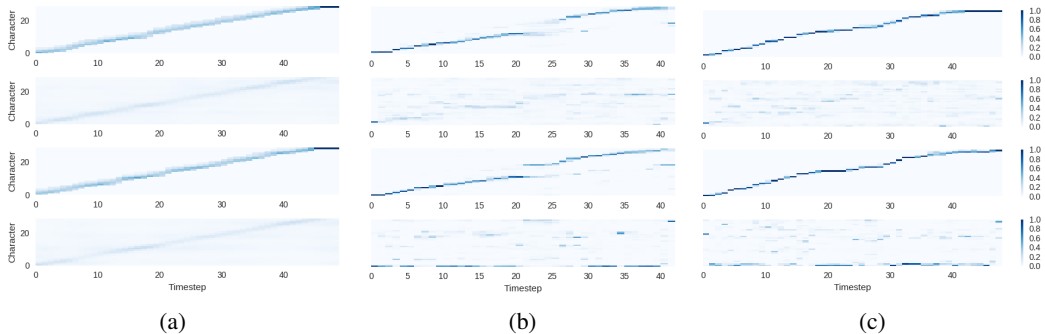

(a)                                    (b)                                    (c)

Figure 4: Attention distributions (a) before training, (b) after training, but without inference constraints, (c) with inference constraints applied to the first and third layers. (We empirically observe that fixing the attention of one or two dominant layers is sufficient for high-quality output.)

be fixed for the key to the ratio of output timesteps to input timesteps (computed across the entire dataset). For multi-speaker datasets, $\omega_s$ is computed for both the key and query from the speaker embedding for each speaker (depicted in Fig. 3). As sine and cosine functions form an orthonormal basis, this initialization yields an attention distribution in the form of a diagonal line (see Fig. 4 (a)). We initialize the fully-connected layer weights used to compute hidden attention vectors to the same values for the query projection and the key projection. Positional encodings are used in all attention blocks. We use context normalization as in Gehring et al. (2017). A fully-connected layer is applied to the context vector to generate the output of the attention block. Overall, positional encodings improve the convolutional attention mechanism.

Production-quality TTS systems have very low tolerance for attention errors. Hence, besides positional encodings, we consider additional strategies to eliminate the cases of repeating or skipping words. One approach is to substitute the canonical attention mechanism with the monotonic attention mechanism introduced in Raffel et al. (2017), which approximates hard-monotonic stochastic decoding with soft-monotonic attention by training in expectation.[7] Despite the improved monotonicity, this strategy may yield a more diffused attention distribution. In some cases, several char-

---

[7]The paper Raffel et al. (2017) also proposes hard monotonic attention process by sampling. It aims to improve the inference speed by only attending over states that are selected via sampling, and thus avoiding compute over future states. In our work, we did not benefit from such speedup, and we observed poor attention behavior in some cases, e.g. being stuck on the first or last character.

acters are attended at the same time and high quality speech couldn't be obtained. We attribute this to the unnormalized attention coefficients of the soft alignment, potentially resulting in weak signal from the encoder. Thus, we propose an alternative strategy of constraining attention weights only at inference to be monotonic, preserving the training procedure without any constraints. Instead of computing the softmax over the entire input, we instead compute the softmax only over a fixed window starting at the last attended-to position and going forward several timesteps [8]. The initial position is set to zero and is later computed as the index of the highest attention weight within the current window. This strategy also enforces monotonic attention at inference as shown in Fig. 4, and yields superior speech quality.

## 3.7 CONVERTER

The converter network takes as inputs the activations from the last hidden layer of the decoder, applies several non-causal convolution blocks, and then predicts parameters for downstream vocoders. Unlike the decoder, the converter is non-causal and non-autoregressive, so it can use future context from the decoder to predict its outputs.

The loss function of the converter network depends on the type of the vocoder used:

1. **Griffin-Lim vocoder**: Griffin-Lim algorithm converts spectrograms to time-domain audio waveforms by iteratively estimating the unknown phases. We find raising the spectrogram to a power parametrized by a *sharpening factor* before waveform synthesis is helpful for improved audio quality, as suggested in Wang et al. (2017). L1 loss is used for prediction of linear-scale log-magnitude spectrograms.

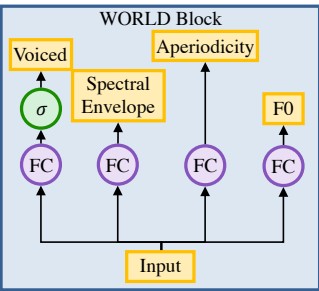

Figure 5: Generated WORLD vocoder parameters with fully connected (FC) layers.

2. **WORLD vocoder**: The WORLD vocoder is based on (Morise et al., 2016). As vocoder parameters, we predict a boolean value (whether the current frame is voiced or unvoiced), an F0 value (if the frame is voiced), the spectral envelope, and the aperiodicity parameters. We use a cross-entropy loss for the voiced-unvoiced prediction, and L1 losses for all other predictions (see Fig. 5),

3. **WaveNet vocoder**: We separately train a WaveNet to be used as a vocoder treating mel-scale log-magnitude spectrograms as vocoder parameters. These vocoder parameters are input as external conditioners to the network. The WaveNet is trained using ground-truth mel-spectragrams and audio waveforms. The architecture besides the conditioner is similar to the WaveNet described in Arık et al. (2017). While the WaveNet in Arık et al. (2017) is conditioned with linear-scale log-magnitude spectrograms, we observed better performance with mel-scale spectrograms, which corresponds to a more compact representation of audio. In addition to L1 loss on mel-scale spectrograms at decode, L1 loss on linear-scale spectrogram is also applied as Griffin-Lim vocoder.

---

[8]We use a window size of 3 in our experiments.

| Text Input | Attention | Inference constraint | Repeat | Mispronounce | Skip |
|---|---|---|---|---|---|
| Characters-only | Dot-Product | Yes | 3 | 35 | 19 |
| Phonemes & Characters | Dot-Product | No | 12 | 10 | 15 |
| **Phonemes & Characters** | **Dot-Product** | **Yes** | **1** | **4** | **3** |
| Phonemes & Characters | Monotonic | No | 5 | 9 | 11 |

Table 1: Attention error counts for single-speaker Deep Voice 3 models on the 100-sentence test set, given in Appendix E. One or more mispronunciations, skips, and repeats count as a single mistake per utterance. "Phonemes & Characters" refers to the model trained with a joint character and phoneme representation, as discussed in Section 3.2. We did not include phoneme-only models because the test set contains out-of-vocabulary words. All models use Griffin-Lim as their vocoder.

## 4 RESULTS

In this section, we present several different experiments and metrics to evaluate our speech synthesis system. We quantify the performance of our system and compare it to other recently published neural TTS systems.

**Data:** For single-speaker synthesis, we use an internal English speech dataset containing approximately 20 hours of audio with a sample rate of 48 kHz. For multi-speaker synthesis, we use the VCTK (Yamagishi et al., 2009) and LibriSpeech (Panayotov et al., 2015) datasets. The VCTK dataset consists of audios for 108 speakers, with a total duration of ∼44 hours. The LibriSpeech dataset consists of audios for 2484 speakers, with a total duration of ∼820 hours. The sample rate is 48 kHz for VCTK and 16 kHz for LibriSpeech.

**Fast Training:** We compare Deep Voice 3 to Tacotron, a recently published attention-based TTS system. For our system on single-speaker data, the average training iteration time (for batch size 4) is 0.06 seconds using one GPU as opposed to 0.59 seconds for Tacotron, indicating a ten-fold increase in training speed. In addition, Deep Voice 3 converges after ∼ 500K iterations for all three datasets in our experiment, while Tacotron requires ∼ 2M iterations as suggested in Wang et al. (2017). This significant speedup is due to the fully-convolutional architecture of Deep Voice 3, which exploits the parallelism of a GPU during training.

**Attention Error Modes:** Attention-based neural TTS systems may run into several error modes that can reduce synthesis quality – including (i) repeated words, (ii) mispronunciations, and (iii) skipped words. [9] One reason for (i) and (iii) is that the attention-based model does not impose a monotonically progressing mechanism. In order to track the occurrence of attention errors, we construct a custom 100-sentence test set (see Appendix E) that includes particularly-challenging cases from deployed TTS systems (e.g. dates, acronyms, URLs, repeated words, proper nouns, foreign words etc.) Attention error counts are listed in Table 1 and indicate that the model with joint representation of characters and phonemes, trained with standard attention mechanism but enforced the monotonic constraint at inference, largely outperforms other approaches.

**Naturalness:** We demonstrate that choice of waveform synthesis matters for naturalness ratings and compare it to other published neural TTS systems. Results in Table 2 indicate that WaveNet, a neural vocoder, achieves the highest MOS of 3.78, followed by WORLD and Griffin-Lim at 3.63 and 3.62, respectively. Thus, we show that the most natural waveform synthesis can be done with a neural vocoder, and that basic spectrogram inversion techniques can match advanced vocoders with high quality single speaker data. The WaveNet vocoder sounds more natural as the WORLD vocoder introduces various noticeable artifacts. Yet, lower inference latency may render the WORLD vocoder preferable: the heavily engineered WaveNet implementation runs at 3X realtime per CPU core (Arık et al., 2017), while WORLD runs up to 40X realtime per CPU core (see the subsection below).

---

[9] As an example, consider the phrase "DOMINANT VEGETARIAN", which should be pronounced with phonemes "D AA M AH N AH N T . V EH JH AH T EH R IY AH N ." The following are example errors for the above three error modes:
(i) "D AA M AH N AH N T . V EH JH AH T EH T EH R IY AH N .",
(ii) "D AE M AH N AE N T . V EH JH AH T EH R IY AH N .",
(iii) "D AH N T . V EH JH AH T EH R IY AH N ."

| Model | Mean Opinion Score (MOS) |
|---|---|
| Deep Voice 3 (Griffin-Lim) | $3.62 \pm 0.31$ |
| Deep Voice 3 (WORLD) | $3.63 \pm 0.27$ |
| Deep Voice 3 (WaveNet) | $3.78 \pm 0.30$ |
| Tacotron (WaveNet) | $3.78 \pm 0.34$ |
| Deep Voice 2 (WaveNet) | $2.74 \pm 0.35$ |

Table 2: Mean Opinion Score (MOS) ratings with 95% confidence intervals using different waveform synthesis methods. We use the crowdMOS toolkit (Ribeiro et al., 2011); batches of samples from these models were presented to raters on Mechanical Turk. Since batches contained samples from all models, the experiment naturally induces a comparison between the models.

| Model | MOS (VCTK) | MOS (LibriSpeech) |
|---|---|---|
| Deep Voice 3 (Griffin-Lim) | $3.01 \pm 0.29$ | $2.37 \pm 0.24$ |
| Deep Voice 3 (WORLD) | $3.44 \pm 0.32$ | $2.89 \pm 0.38$ |
| Deep Voice 2 (WaveNet) | $3.69 \pm 0.23$ | - |
| Tacotron (Griffin-Lim) | $2.07 \pm 0.31$ | - |
| Ground truth | $4.69 \pm 0.04$ | $4.51 \pm 0.18$ |

Table 3: MOS ratings with 95% confidence intervals for audio clips from neural TTS systems on multi-speaker datasets. We also use crowdMOS toolkit; batches of samples including ground truth were presented to human raters. Multi-speaker Tacotron implementation and hyperparameters are based on Arık et al. (2017), which is a proof-of-concept implementation. Deep Voice 2 and Tacotron systems were not trained for the LibriSpeech dataset due to prohibitively long time required to optimize hyperparameters.

**Multi-Speaker Synthesis:** To demonstrate that our model is capable of handling multi-speaker speech synthesis effectively, we train our models on the VCTK and LibriSpeech data sets. For LibriSpeech (an ASR dataset), we apply a preprocessing step of standard denoising (using SoX (Bagwell, 2017)) and splitting long utterances into multiple at pause locations (which are determined by Gentle (Ochshorn & Hawkins, 2017)). Results are presented in Table 3. We purposefully include ground-truth samples in the set being evaluated, because the accents in datasets are likely to be unfamiliar to our North American crowdsourced raters. Our model with the WORLD vocoder achieves a comparable MOS of 3.44 on VCTK in contrast to 3.69 from Deep Voice 2, which is the state-of-the-art multi-speaker neural TTS system using WaveNet as vocoder and seperately optimized phoneme duration and fundamental frequency prediction models. We expect further improvement by using WaveNet for multi-speaker synthesis, although it may substantially slow down inference. The MOS on LibriSpeech is lower compared to VCTK, which we mainly attribute to the lower quality of the training dataset due to the various recording conditions and noticeable background noise. [10] In the literature, Yamagishi et al. (2010) also observes worse performance, when apply parametric TTS method to different ASR datasets with hundreds of speakers. Lastly, we find that the learned speaker embeddings lie in a meaningful latent space (see Fig. 7 in Appendix D).

**Optimizing Inference for Deployment:** In order to deploy a neural TTS system in a cost-effective manner, the system must be able to handle as much traffic as alternative systems on a comparable amount of hardware. To do so, we target a throughput of ten million queries per day or 116 queries per second (QPS) [11] on a single-GPU server with twenty CPU cores, which we find is comparable in cost to commercially deployed TTS systems. By implementing custom GPU kernels for the Deep Voice 3 architecture and parallelizing WORLD synthesis across CPUs, we demonstrate that our model can handle ten million queries per day. We provide more details on the implementation in Appendix B.

---

[10] We test Deep Voice 3 on a subsampled LibriSpeech with only 108 speakers (same as VCTK) and observe worse quality of generated samples than VCTK.

[11] A query is defined as synthesizing the audio for a one second utterance.

## 5    CONCLUSION

We introduce Deep Voice 3, a neural text-to-speech system based on a novel fully-convolutional sequence-to-sequence acoustic model with a position-augmented attention mechanism. We describe common error modes in sequence-to-sequence speech synthesis models and show that we successfully avoid these common error modes with Deep Voice 3. We show that our model is agnostic of the waveform synthesis method, and adapt it for Griffin-Lim spectrogram inversion, WaveNet, and WORLD vocoder synthesis. We demonstrate also that our architecture is capable of multispeaker speech synthesis by augmenting it with trainable speaker embeddings, a technique described in Deep Voice 2. Finally, we describe the production-ready Deep Voice 3 system in full including text normalization and performance characteristics, and demonstrate state-of-the-art quality through extensive MOS evaluations. Future work will involve improving the implicitly learned grapheme-to-phoneme model, jointly training with a neural vocoder, and training on cleaner and larger datasets to scale to model the full variability of human voices and accents from hundreds of thousands of speakers.

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

# Appendices

## A DETAILED MODEL ARCHITECTURE OF DEEP VOICE 3

The detailed model architecture in depicted in Fig. 6.

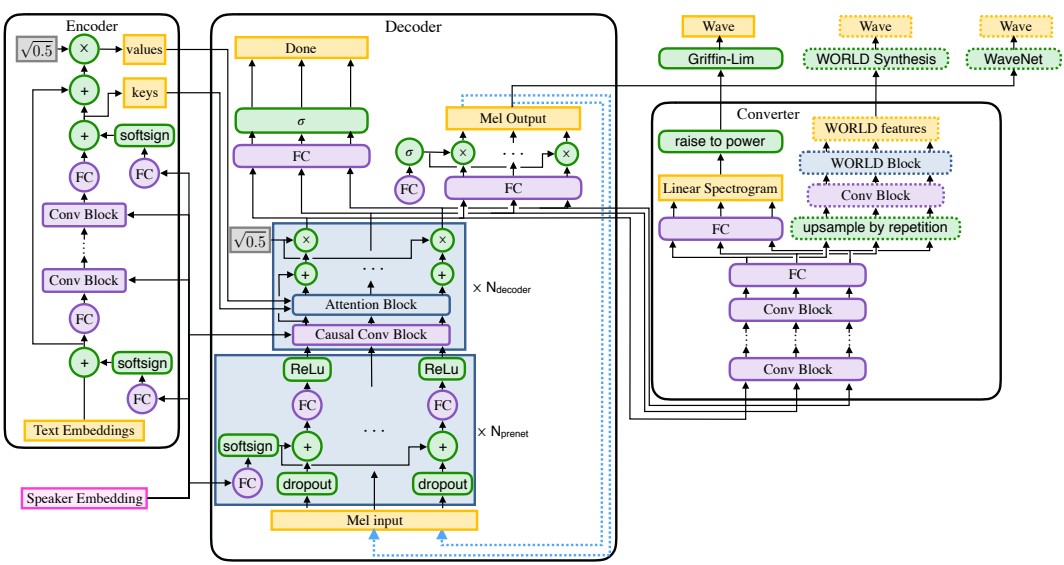

Figure 6: Deep Voice 3 uses a deep residual convolutional network to encode text and/or phonemes into per-timestep *key* and *value* vectors for an attentional decoder. The decoder uses these to predict the mel-band log magnitude spectrograms that correspond to the output audio. (Light blue dotted arrows depict the autoregressive synthesis process during inference.) The hidden state of the decoder then gets fed to a converter network to output linear spectrograms for Griffin-Lim or parameters for WORLD, which can be used to synthesize the final waveform. Weight normalization (Salimans & Kingma, 2016) is applied to all convolution filters and fully-connected layer weight matrices in the model.

## B OPTIMIZING DEEP VOICE 3 FOR DEPLOYMENT

Running inference with a TensorFlow graph turns out to be prohibitively expensive, averaging approximately 1 QPS [12]. Instead, we implement custom GPU kernels for Deep Voice 3 inference. Due to the complexity of the model and the large number of output timesteps, launching individual kernels for different operations in the graph (convolutions, matrix multiplications, unary and binary operations etc.) is impractical: the overhead of launch a CUDA kernel is approximately 50 $\mu$s, which, when aggregated across all operations in the model and all output timesteps, limits throughput to approximately 10 QPS. Thus, we implement a single kernel for the entire model, which avoids the overhead of launching many CUDA kernels. Finally, instead of batching computation in the kernel, our kernel operates on a single utterance and we launch as many concurrent streams as there are Streaming Multiprocessors (SMs) on the GPU. Every kernel is launched with one block, so we expect the GPU to schedule one block per SM, allowing us to scale inference speed linearly with the number of SMs.

On a single Nvidia Tesla P100 GPU with 56 SMs, we achieve an inference speed of 115 QPS, which corresponds to our target ten million queries per day. We parallelize WORLD synthesis across all 20 CPUs on the server, permanently pinning threads to CPUs in order to maximize cache performance.

---

[12]The poor TensorFlow performance is due to the overhead of running the graph evaluator over hundreds of nodes and hundreds of timesteps. Using a technology such as XLA with TensorFlow could speed up evaluation but is unlikely to match the performance of a hand-written kernel.

In this setup, GPU inference is the bottleneck, as WORLD synthesis on 20 cores is faster than 115 QPS.

We believe that inference can be made significantly faster through more optimized kernels, smaller models, and fixed-precision arithmetic; we leave these aspects to future work.

## C    MODEL HYPERPARAMETERS

All hyperparameters of the models used in this paper are shown in Table 4.

| Parameter | Single-Speaker | VCTK | LibriSpeech |
|---|---|---|---|
| FFT Size | 4096 | 4096 | 4096 |
| FFT Window Size / Shift | 2400 / 600 | 2400 / 600 | 1600 / 400 |
| Audio Sample Rate | 48000 | 48000 | 16000 |
| Reduction Factor $r$ | 4 | 4 | 4 |
| Mel Bands | 80 | 80 | 80 |
| Sharpening Factor | 1.4 | 1.4 | 1.4 |
| Character Embedding Dim. | 256 | 256 | 256 |
| Encoder Layers / Conv. Width / Channels | 7 / 5 / 64 | 7 / 5 / 128 | 7 / 5 / 256 |
| Decoder Affine Size | 128, 256 | 128, 256 | 128, 256 |
| Decoder Layers / Conv. Width | 4 / 5 | 6 / 5 | 8 / 5 |
| Attention Hidden Size | 128 | 256 | 256 |
| Position Weight / Initial Rate | 1.0 / 6.3 | 0.1 / 7.6 | 0.1 / 2.6 |
| Converter Layers / Conv. Width / Channels | 5 / 5 / 256 | 6 / 5 / 256 | 8 / 5 / 256 |
| Dropout Keep Probability | 0.95 | 0.95 | 0.99 |
| Number of Speakers | 1 | 108 | 2484 |
| Speaker Embedding Dim. | - | 16 | 512 |
| ADAM Learning Rate | 0.001 | 0.0005 | 0.0005 |
| Anneal Rate / Anneal Interval | - | 0.98 / 30000 | 0.95 / 30000 |
| Batch Size | 16 | 16 | 16 |
| Max Gradient Norm | 100 | 100 | 50.0 |
| Gradient Clipping Max. Value | 5 | 5 | 5 |

Table 4: Hyperparameters used for best models for the three datasets used in the paper.

## D    LATENT SPACE OF THE LEARNED EMBEDDINGS

Similar to Arık et al. (2017), we apply principal component analysis to the learned speaker embeddings and analyze the speakers based on their ground truth genders. Fig. 7 shows the genders of the speakers in the space spanned by the first two principal components. We observe a very clear separation between male and female genders, suggesting the low-dimensional speaker embeddings constitute a meaningful latent space.

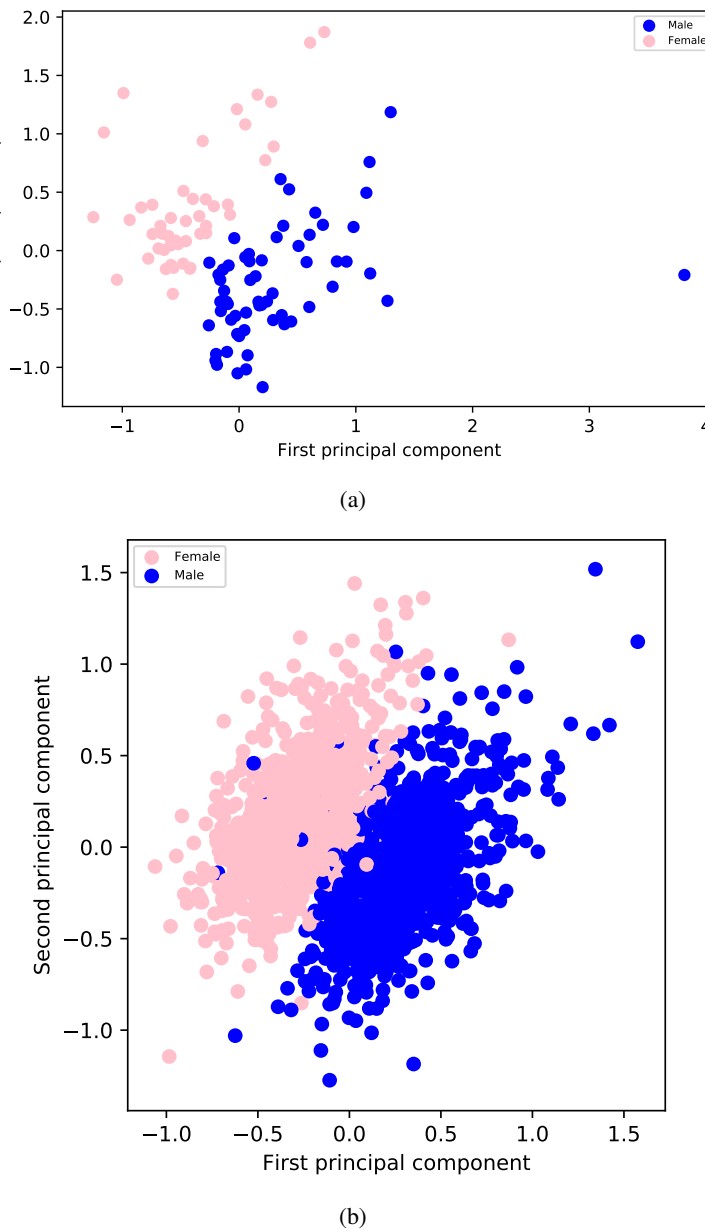

Figure 7: The first two principal components of the learned embeddings for (a) VCTK dataset (108 speakers) and (b) LibriSpeech dataset (2484 speakers).

## E    100-SENTENCE TEST SET

The 100 sentences used to quantify the results in Table 1 are listed below (note that % symbol corresponds to pause):

1. A B C%.
2. X Y Z%.
3. HURRY%.
4. WAREHOUSE%.
5. REFERENDUM%.
6. IS IT FREE%?
7. JUSTIFIABLE%.
8. ENVIRONMENT%.
9. A DEBT RUNS%.
10. GRAVITATIONAL%.
11. CARDBOARD FILM%.
12. PERSON THINKING%.
13. PREPARED KILLER%.
14. AIRCRAFT TORTURE%.
15. ALLERGIC TROUSER%.
16. STRATEGIC CONDUCT%.
17. WORRYING LITERATURE%.
18. CHRISTMAS IS COMING%.
19. A PET DILEMMA THINKS%.
20. HOW WAS THE MATH TEST%?
21. GOOD TO THE LAST DROP%.
22. AN M B A AGENT LISTENS%.
23. A COMPROMISE DISAPPEARS%.
24. AN AXIS OF X Y OR Z FREEZES%.
25. SHE DID HER BEST TO HELP HIM%.
26. A BACKBONE CONTESTS THE CHAOS%.
27. TWO A GREATER THAN TWO N NINE%.
28. DON'T STEP ON THE BROKEN GLASS%.
29. A DAMNED FLIPS INTO THE PATIENT%.
30. A TRADE PURGES WITHIN THE B B C%.
31. I'D RATHER BE A BIRD THAN A FISH%.
32. I HEAR THAT NANCY IS VERY PRETTY%.
33. I WANT MORE DETAILED INFORMATION%.
34. PLEASE WAIT OUTSIDE OF THE HOUSE%.
35. N A S A EXPOSURE TUNES THE WAFFLE%.
36. A MIST DICTATES WITHIN THE MONSTER%.
37. A SKETCH ROPES THE MIDDLE CEREMONY%.
38. EVERY FAREWELL EXPLODES THE CAREER%.
39. SHE FOLDED HER HANDKERCHIEF NEATLY%.
40. AGAINST THE STEAM CHOOSES THE STUDIO%.
41. ROCK MUSIC APPROACHES AT HIGH VELOCITY%.
42. NINE ADAM BAYE STUDY ON THE TWO PIECES%.
43. AN UNFRIENDLY DECAY CONVEYS THE OUTCOME%.
44. ABSTRACTION IS OFTEN ONE FLOOR ABOVE YOU%.
45. A PLAYED LADY RANKS ANY PUBLICIZED PREVIEW%.
46. HE TOLD US A VERY EXCITING ADVENTURE STORY%.
47. ON AUGUST TWENTY EIGTH%MARY PLAYS THE PIANO%.
48. INTO A CONTROLLER BEAMS A CONCRETE TERRORIST%.
49. I OFTEN SEE THE TIME ELEVEN ELEVEN ON CLOCKS%.
50. IT WAS GETTING DARK%AND WE WEREN'T THERE YET%.
51. AGAINST EVERY RHYME STARVES A CHORAL APPARATUS%.
52. EVERYONE WAS BUSY%SO I WENT TO THE MOVIE ALONE%.
53. I CHECKED TO MAKE SURE THAT HE WAS STILL ALIVE%.
54. A DOMINANT VEGETARIAN SHIES AWAY FROM THE G O P%.
55. JOE MADE THE SUGAR COOKIES%SUSAN DECORATED THEM%.
56. I WANT TO BUY A ONESIE%BUT KNOW IT WON'T SUIT ME%.
57. A FORMER OVERRIDE OF Q W E R T Y OUTSIDE THE POPE%.
58. F B I SAYS THAT C I A SAYS%I'LL STAY AWAY FROM IT%.
59. ANY CLIMBING DISH LISTENS TO A CUMBERSOME FORMULA%.
60. SHE WROTE HIM A LONG LETTER%BUT HE DIDN'T READ IT%.
61. DEAR%BEAUTY IS IN THE HEAT NOT PHYSICAL%I LOVE YOU%.
62. AN APPEAL ON JANUARY FIFTH DUPLICATES A SHARP QUEEN%.
63. A FAREWELL SOLOS ON MARCH TWENTY THIRD SHAKES NORTH%.
64. HE RAN OUT OF MONEY%SO HE HAD TO STOP PLAYING POKER%.

65. FOR EXAMPLE%A NEWSPAPER HAS ONLY REGIONAL DISTRIBUTION T%.
66. I CURRENTLY HAVE FOUR WINDOWS OPEN UP%AND I DON'T KNOW WHY%.
67. NEXT TO MY INDIRECT VOCAL DECLINES EVERY UNBEARABLE ACADEMIC%.
68. OPPOSITE HER SOUNDING BAG IS A M C'S CONFIGURED THOROUGHFARE%.
69. FROM APRIL EIGHTH TO THE PRESENT%I ONLY SMOKE FOUR CIGARETTES%.
70. I WILL NEVER BE THIS YOUNG AGAIN%EVER%OH DAMN%I JUST GOT OLDER%.
71. A GENEROUS CONTINUUM OF AMAZON DOT COM IS THE CONFLICTING WORKER%.
72. SHE ADVISED HIM TO COME BACK AT ONCE%THE WIFE LECTURES THE BLAST%.
73. A SONG CAN MAKE OR RUIN A PERSON'S DAY IF THEY LET IT GET TO THEM%.
74. SHE DID NOT CHEAT ON THE TEST%FOR IT WAS NOT THE RIGHT THING TO DO%.
75. HE SAID HE WAS NOT THERE YESTERDAY%HOWEVER%MANY PEOPLE SAW HIM THERE%.
76. SHOULD WE START CLASS NOW%OR SHOULD WE WAIT FOR EVERYONE TO GET HERE%?
77. IF PURPLE PEOPLE EATERS ARE REAL%WHERE DO THEY FIND PURPLE PEOPLE TO EAT%?
78. ON NOVEMBER EIGHTEENTH EIGHTEEN TWENTY ONE%A GLITTERING GEM IS NOT ENOUGH%.
79. A ROCKET FROM SPACE X INTERACTS WITH THE INDIVIDUAL BENEATH THE SOFT FLAW%.
80. MALLS ARE GREAT PLACES TO SHOP%I CAN FIND EVERYTHING I NEED UNDER ONE ROOF%.
81. I THINK I WILL BUY THE RED CAR%OR I WILL LEASE THE BLUE ONE%THE FAITH NESTS%.
82. ITALY IS MY FAVORITE COUNTRY%IN FACT%I PLAN TO SPEND TWO WEEKS THERE NEXT YEAR%.
83. I WOULD HAVE GOTTEN W W W DOT GOOGLE DOT COM%BUT MY ATTENDANCE WASN'T GOOD ENOUGH%.
84. NINETEEN TWENTY IS WHEN WE ARE UNIQUE TOGETHER UNTIL WE REALISE%WE ARE ALL THE SAME%.
85. MY MUM TRIES TO BE COOL BY SAYING H T T P COLON SLASH SLASH W W W B A I D U DOT COM%.
86. HE TURNED IN THE RESEARCH PAPER ON FRIDAY%OTHERWISE%HE EMAILED A S D F AT YAHOO DOT ORG%.
87. SHE WORKS TWO JOBS TO MAKE ENDS MEET%AT LEAST%THAT WAS HER REASON FOR NOT HAVING TIME TO JOIN US%.
88. A REMARKABLE WELL PROMOTES THE ALPHABET INTO THE ADJUSTED LUCK%THE DRESS DODGES ACROSS MY ASSAULT%.
89. A B C D E F G H I J K L M N O P Q R S T U V W X Y Z ONE TWO THREE FOUR FIVE SIX SEVEN EIGHT NINE TEN%.
90. ACROSS THE WASTE PERSISTS THE WRONG PACIFIER%THE WASHED PASSENGER PARADES UNDER THE INCORRECT COMPUTER%.
91. IF THE EASTER BUNNY AND THE TOOTH FAIRY HAD BABIES WOULD THEY TAKE YOUR TEETH AND LEAVE CHOCOLATE FOR YOU%?
92. SOMETIMES%ALL YOU NEED TO DO IS COMPLETELY MAKE AN ASS OF YOURSELF AND LAUGH IT OFF TO REALISE THAT LIFE ISN'T SO BAD AFTER ALL%.
93. SHE BORROWED THE BOOK FROM HIM MANY YEARS AGO AND HASN'T YET RETURNED IT%WHY WON'T THE DISTINGUISHING LOVE JUMP WITH THE JUVENILE%?
94. LAST FRIDAY IN THREE WEEK'S TIME I SAW A SPOTTED STRIPED BLUE WORM SHAKE HANDS WITH A LEGLESS LIZARD%THE LAKE IS A LONG WAY FROM HERE%.
95. I WAS VERY PROUD OF MY NICKNAME THROUGHOUT HIGH SCHOOL BUT TODAY%I COULDN'T BE ANY DIFFERENT TO WHAT MY NICKNAME WAS%THE METAL LUSTS%THE RANGING CAPTAIN CHARTERS THE LINK%.
96. I AM HAPPY TO TAKE YOUR DONATION%ANY AMOUNT WILL BE GREATLY APPRECIATED%THE WAVES WERE CRASHING ON THE SHORE%IT WAS A LOVELY SIGHT%THE PARADOX STICKS THIS BOWL ON TOP OF A SPONTANEOUS TEA%.
97. A PURPLE PIG AND A GREEN DONKEY FLEW A KITE IN THE MIDDLE OF THE NIGHT AND ENDED UP SUNBURNT%THE CONTAINED ERROR POSES AS A LOGICAL TARGET%THE DIVORCE ATTACKS NEAR A MISSING DOOM%THE OPERA FINES THE DAILY EXAMINER INTO A MURDERER%.
98. AS THE MOST FAMOUS SINGLER-SONGWRITER%JAY CHOU GAVE A PERFECT PERFORMANCE IN BEIJING ON MAY TWENTY FOURTH%TWENTY FIFTH%AND TWENTY SIXTH TWENTY THREE ALL THE FANS THOUGHT HIGHLY OF HIM AND TOOK PRIDE IN HIM ALL THE TICKETS WERE SOLD OUT%.
99. IF YOU LIKE TUNA AND TOMATO SAUCE%TRY COMBINING THE TWO%IT'S REALLY NOT AS BAD AS IT SOUNDS%THE BODY MAY PERHAPS COMPENSATES FOR THE LOSS OF A TRUE METAPHYSICS%THE CLOCK WITHIN THIS BLOG AND THE CLOCK ON MY LAPTOP ARE ONE HOUR DIFFERENT FROM EACH OTHER%.
100. SOMEONE I KNOW RECENTLY COMBINED MAPLE SYRUP AND BUTTERED POPCORN THINKING IT WOULD TASTE LIKE CARAMEL POPCORN%IT DIDN'T AND THEY DON'T RECOMMEND ANYONE ELSE DO IT EITHER%THE GENTLEMAN MARCHES AROUND THE PRINCIPAL%THE DIVORCE ATTACKS NEAR A MISSING DOOM%THE COLOR MISPRINTS A CIRCULAR WORRY ACROSS THE CONTROVERSY%.

