# OpenReview forum: "Deep Voice 3: Scaling Text-to-Speech with Convolutional Sequence Learning"
_ICLR.cc/2018/Conference — Accept (Poster)_

### Official Review · AnonReviewer1 · 2017-11-26
**Impressive results, but missing some details and insights**

**Rating:** 6
**Confidence:** 5

**Review:**

This paper discusses a text-to-speech system which is based on a convolutional attentive seq2seq architecture.  It covers experiments on a few datasets, testing the model's ability to handle increasing numbers of speakers.

By and large, this is a "system" paper - it mostly describes the successful application of many different existing ideas to an important problem (with some exceptions, e.g. the novel method of enforcing monotonic alignments during inference).  In this type of paper, I typically am most interested in hearing about *why* a particular design choice was made, what alternatives were tried, and how different ideas worked.  This paper is lacking in this regard - I frequently was left looking for more insight into the particular system that was designed.  Beyond that, I think more detailed description of the system would be necessary in order to reimplement it suitably (another important potential takeaway for a "system" paper).  Separately, I the thousands-of-speakers results are just not that impressive - a MOS of 2 is not really useable in the real-world.  For that reason, I think it's a bit disingenuous to sell this system as "2000-Speaker Neural Text-to-Speech".

For the above reasons, I'm giving the paper a "marginally above" rating.  If the authors provide improved insight, discussion of system specifics, and experiments, I'd be open to raising my review.  Below, I give some specific questions and suggestions that could be addressed in future drafts.

- It might be worth giving a sentence or two defining the TTS problem - the paper is written assuming background knowledge about the problem setting, including different possible input sources, what a vocoder is, etc.  The ICLR community at large may not have this domain-specific knowledge.
- Why "softsign" and not tanh?  Seems like an unusual choice.
- What do the "c" and "2c" in Figure 2a denote?
- Why scale (h_k + h_e) by \sqrt{0.5} when computing the attention value vectors?
- "An L1 loss is computed using the output spectrograms" I assume you mean the predicted and target spectrograms are compared via an L1 loss.  Why L1?
- In Vaswani et al., it was shown that a learned positional encoding worked about as well as the sinusoidal position encodings despite being potentially more flexible/less "hand-designed" for machine translation.  Did you also try this for TTS?  Any insight?
- Some questions about monotonic attention: Did you use the training-time "soft" monotonic attention algorithm from Raffel et al. during training and inference, or did you use the "hard" monotonic attention at inference time?  IIUC the "soft" algorithm doesn't actually force strict monotonicity.  You wrote "monotonic attention results in the model frequently mumbling words", can you provide evidence/examples of this?  Why do you think this happens?  The monotonic attention approach seems more principled than post-hoc limiting softmax attention to be monotonic, why do you think it didn't work as well?
- I can't find an actual reference to what you mean by a "wavenet vocoder".  The original wavenet paper describes an autoregressive model for waveform generation.  In order to use it as a vocoder, you'd have to do conditioning in some way.  How?  What was the structure of the wavenet you used?   Why?  These details appear to be missing.  All you write is the sentence (which seems to end without a period) "In the WaveNet vocoder, we use mel-scale spectrograms from the decoder to condition a Wavenet, which was trained separated".
- Can you provide examples of the mispronunciations etc. which were measured for Table 1?  Was the evaluation of each attention mechanism done blindly?
- The 2.07 MOS figure produced for tacotron seems extremely low, and seems to indicate that something went wrong or that insufficient care was taken to report this baseline.  How did you adapt tacotron (which as I understand is a single-speaker model) to the multi-speaker setting?
- Table 3 begs the question of whether Deep Voice 3 can outperform Deep Voice 2 when using a wavenet vocoder on VCTK (or improve upon the poor 2.09 MOS score reported).  Why wasn't this experiment run?
- The paragraph and appendix about deploying at scale is interesting and impressive, but seems a bit out of place - it probably makes more sense to include this information in a separate "systems" paper.

---

> ### Author Response · Authors · 2018-01-05
> **Thank you for your in-depth review, the comments are very helpful in improving the quality of our paper.**
>
> As another reviewer also raised a similar concern, we have paid significant attention to improve the explanations to motivate the model architecture choices in this paper.  For example:
> 1) We have added a new section ("Convolution Blocks for Sequential Processing") to motivate the architecture design choices for the convolution blocks used in our model.
> 2) In the "Encoder" section, we have added the motivation behind mixing key vector h_k and embedding h_e. (This is in regards to your question about mixing h_k and h_e.)
> 3) In the "Decoder" section, we have expanded the explanation of query generation for attention and explain the motivation to use L1 loss.
> 4) In the "Attention Block" section, we have added more explanations for our attention mechanism choices, attention's role in the overall architecture, the choice of positional encodings, and techniques to minimize attention errors.
> 5) In the "Converters" section, we have added clarification and justification for the relationship between the decoder hidden state and the converter/vocoder.
>
>  We note that due to the required additions by the Reviewers, our page limit has exceeded the suggested.
>
>  - "Separately, the thousands-of-speakers results are just not that impressive - a MOS of 2 is not really useable in the real-world ... "
> * Since this submission, we have worked on optimizing the hyperparameters further. We were able to improve MOS from 2.09 to 2.37 by increasing the dimensionality of the speaker embedding and to 2.89 with the WORLD vocoder. We have updated our draft to reflect the improvement. In addition, it should be noted that LibriSpeech is a dataset for automatic speech recognition (ASR), which is recorded in various environments and often contains noticeable background noise. This characteristic is helpful for the robustness of an ASR system, but is harmful for a TTS system. In the literature, Yamagishi et al. (2010) built TTS systems using several ASR corporas with much fewer speakers, and the highest MOS 2.8 is on WSJ dataset which is "cleaner" than Librispeech. We expect a higher MOS score with a "TTS-quality" dataset that is at the scale of LibriSpeech, but it is very expensive to collect.  Also, we are considering to change the title to "Deep Voice 3: Scaling Text-to-Speech with Convolutional Sequence Learning" in the final version.
>
> - “It might be worth giving a sentence or two defining the TTS problem ... ”
> * We have modified the first paragraph to introduce definitions.
>
>  - "Why "softsign" and not tanh? "
> * Softsign is preferred over tanh, because it has does not saturate as easily as nonlinearities based on exponential function and still yields sufficiently large gradients for large inputs. We have added a description in Section 3.3.
>
> - "What do the "c" and "2c" in Figure 2a denote?"
> * "c" denotes the dimensionality of the input. We have added this clarification to the caption of Fig. 2a.
>
> - "Why scale (h_k + h_e) by \sqrt{0.5} when computing the attention value vectors?"
> * The scaling factor \sqrt{0.5} ensures that we preserve the unit variance early in training. It is explained in footnote 4.
>
> - "An L1 loss is computed using the output spectrograms ... Why L1?"
> * Prediction of spectrograms is treated as a regression problem. We choose L1 loss since it yielded the best results empirically. Other common regression loss functions such as L2 loss may suffer from outlier spectral features (which may correspond to non-speech noise). We have clarified this point in Section 3.5.
>
> - "In Vaswani et al., it was shown that a learned positional encoding ... "
> * We didn't try the learned positional encoding in our system. The benefit of adding the positional encoding is significant only at the beginning of training and we do not expect superior audio quality by simply using different positional encodings. We have added these additional details in the paper.

---

> > ### Author Response · Authors · 2018-01-05
> > **Continued response:**
> >
> >
> > - "Some questions about monotonic attention: ... "
> > * We have modified the paragraph to address these questions:
> > "Production-quality TTS systems have very low tolerance for attention errors. Hence, besides positional encodings, we consider additional strategies to minimize the cases of repeating or skipping words. One approach is to substitute the canonical attention mechanism with the monotonic attention mechanism introduced in Raffel et al. (2017), which approximates hard-monotonic stochastic decoding with soft-monotonic attention by training in expectation.\footnote{The paper Raffel et al. (2017) also proposes hard monotonic attention process by sampling. It aims to improve the inference speed by only attending over states that are selected via sampling, and thus avoiding compute over future states. In our work, we did not benefit from such speedup, and we observed poor attention behavior in some cases, e.g. being stuck on the first or last character.} Despite the improved monotonicity, this strategy may yield a more diffused attention distribution. In some cases, several characters are attended at the same time and high quality speech couldn't be obtained. We attribute this to the unnormalized attention coefficients of the soft alignment, potentially resulting in weak signal from the encoder. Thus, we propose an alternative strategy of constraining attention weights only at inference to be monotonic, preserving the training procedure without any constraints. Instead of computing the softmax over the entire input, we instead compute the softmax only over a fixed window starting at the last attended-to position and going forward several timesteps~\footnote{We use a window size of 3 in our experiments.}. The initial position is set to zero and is later computed as the index of the highest attention weight within the current window. This strategy also enforces monotonic attention distribution at inference, as shown in Fig. 4 and yields superior speech quality. "
> >
> > - "I can't find an actual reference to what you mean by a "wavenet vocoder" ... "
> > * We have added the following paragraph and footnotes for clarification:
> > "We separately train a WaveNet to be used as a vocoder treating mel-scale log-magnitude spectrograms as vocoder parameters. These vocoder parameters are input as external conditioners to the network. The training procedure and the architecture besides the conditioner are similar to the WaveNet described in Deep Voice 2. While the WaveNet in Deep Voice 2 is conditioned with linear-scale log-magnitude spectrograms, we observed better performance with mel-scale spectrograms, which corresponds to a more compact representation of audio. To predict mel-scale spectrograms, L1 loss on linear-scale spectrograms are also used besides L1 loss on mel-scale spectrograms at decoder. \footnote{A loss on the converter output can be considered in the context of multi-task learning in conjunction with decoder output, since the goal is to improve the estimation accuracy of the mel-scale spectrograms at the converter input.}"
> >
> > - "Can you provide examples of the mispronunciations etc ... "
> > * We add a footnote 11 to exemplify attention errors. Our error evaluation is done blindly.
> >
> >  - "The 2.07 MOS figure produced for tacotron seems extremely low ... How did you adapt tacotron to multi-speaker setting?"
> > * We use the multi-speaker Tacotron described in the Deep Voice 2 paper, which is only a proof-of-concept implementation. We include this result for completeness. We will clarify it in main text.
> >
> > - "Table 3 begs the question of whether Deep Voice 3 can outperform Deep Voice 2 when using a wavenet vocoder ... "
> > * We didn't have sufficient time and resources to perform hyperparameter search for WaveNet on VCTK and LibriSpeech datasets. We leave it for future work.
> >
> >  - "The paragraph and appendix about deploying at scale is interesting and impressive ... a separate "systems" paper."
> > * Thanks for your suggestion! We don't think the contributions for deployment at scale warrants a separate paper, so we prefer to keep the majority of content in the appendix in case other engineers and researchers may benefit from it.

---

### Official Review · AnonReviewer2 · 2017-11-27

**Rating:** 6
**Confidence:** 3

**Review:**

The paper presents a speech synthesis system based on convolution neural networks. The proposed approach is an end-to-end characters to spectrogram system, trained on a very large dataset. The paper also introduces a attention model and can be used with various waveform synthesis methods. The proposed model is shown to match the state -of-the-art approaches performance in speech naturalness.

The paper is clearly written and easy to follow. The relation to previous works is detailed and clear.

The contributions of the paper are significants and an important step towards practical and efficient neural TTS system. The ability to train on a large corpus of speaker 10 times faster than current models is impressive and important for deployment, as is the cost-effective inference and the monotonic attention model.
The experiments on naturalness (Table 2) are convincing and show the viability of the approach. However, the experiments on multi-speaker synthesis (Table 3) are not very strong. The proposed model seems to need to use Wavenet as a vocoder to possibly outperform Deep Voice 2, which will slow down the inference time, one of the strong aspect of the proposed model.

Other comments:

* In Section 2, it is mentioned that RNN-based approaches can leads to attention errors, can the authors elaborate more on that aspect ? It seems important as the proposed approach alleviates these issues, but it is not clear from the paper what these errors are and why they happen.

* In Table 3 there seems to be missing models compared to Table 2, like Tacotron with Wavenet, the authors should explain why in the text.

* The footnote 2 on page 3 looks important enough to be part of the main text.

---

> ### Author Response · Authors · 2018-01-05
> **Thank you for your review; the feedback is very helpful to improve our paper.**
>
> - "The experiments on naturalness (Table 2) are convincing and show the viability of the approach. However, the experiments on multi-speaker synthesis (Table 3) are not very strong ..."
> * Since our submission, we have worked on optimizing the hyperparameters of multi-speaker synthesis further. For Librispeech dataset, we were able to improve MOS to 2.37 by increasing the dimensionality of the speaker embedding and to 2.89 with the WORLD vocoder. We have updated our draft to reflect the improvement.
>
> Other comments:
>
> - "In Section 2, it is mentioned that RNN-based approaches can leads to attention errors, can the authors elaborate more on that aspect? It seems important as the proposed approach alleviates these issues, but it is not clear from the paper what these errors are and why they happen."
> * The attention errors are attributed to canonical attention mechanism rather than the recurrent layers themselves. We have removed that phrase in Section 2 to avoid confusion. We have also added a footnote 11 to exemplify common attention errors.
>
>  - "In Table 3 there seems to be missing models compared to Table 2, like Tacotron with Wavenet, the authors should explain why in the text. "
>  * We have added the reason to the caption of Table 3: "Deep Voice 2 and Tacotron systems were not trained for the LibriSpeech dataset due to the prohibitively long time required to optimize hyperparameters."
>
>  - "The footnote 2 on page 3 looks important enough to be part of the main text."
> * Thanks for your suggestion. We have integrated the footnote into the main text.

---

### Official Review · AnonReviewer3 · 2017-12-04
**Detailed tech report, missing some motivation and comparison experiments**

**Rating:** 7
**Confidence:** 4

**Review:**

This paper provides an overview of the Deep Voice 3 text-to-speech system. It describes the system in a fair amount of detail and discusses some trade-offs w.r.t. audio quality and computational constraints. Some experimental validation of certain architectural choices is also provided.

My main concern with this work is that it reads more like a tech report: it describes the workings and design choices behind one particular system in great detail, but often these choices are simply stated as fact and not really motivated, or compared to alternatives. This makes it difficult to tell which of these aspects are crucial to get good performance, and which are just arbitrary choices that happen to work okay.

As this system was clearly developed with actual deployment in mind (and not purely as an academic pursuit), all of these choices must have been well-deliberated. It is unfortunate that the paper doesn't demonstrate this. I think this makes the work less interesting overall to an ICLR audience. That said, it is perhaps useful to get some insight into what types of models are actually used in practice.

An exception to this is the comparison of "converters", model components that convert the model's internal representation of speech into waveforms. This comparison is particularly interesting because some of the results are remarkable, i.e. Griffin-Lim spectrogram inversion and the WORLD vocoder achieving very similar MOS scores in some cases (Table 2). I wish there would be more of that kind of thing in the paper. The comparison of attention mechanisms is also useful.

I'm on the fence as I think it is nice to get some insight into a practical pipeline which benefits from many current trends in deep learning research (autoregressive models, monotonic attention, ...), but I also feel that the paper is a bit meager when it comes to motivating all the architectural aspects. I think the paper is well written so I've tentatively recommended acceptance.


Other comments:

- The separation of the "decoder" and "converter" stage is not entirely clear to me. It seems that the decoder is trained to predict spectrograms autoregressively, but its final layer is then discarded and its hidden representation is then used as input to the converter stage instead? The motivation for doing this is unclear to me, surely it would be better to train everything end-to-end, including the converter? This seems like an unnecessary detour, what's the reasoning behind this?

- At the bottom of page 2 it is said that "the whole model is trained end-to-end, excluding the vocoder", which I think is an unfortunate turn of phrase. It's either end-to-end, or it isn't.

- In Section 3.3, the point of mixing of h_k and h_e is unclear to me. Why is this done?

- The gated linear unit in Figure 2a shows that speaker embedding information is only injected in the linear part. Has this been experimentally validated to work better than simpler mechanisms such as adding conditioning-dependent biases/gains?

- When the decoder is trained to do autoregressive prediction of spectrograms, is it autoregressive only in time, or also in frequency? I'm guessing it's the former, but this means there is an implicit independence assumption (the intensities in different frequency bins are conditionally independent, given all past timesteps). Has this been taken into consideration? Maybe it doesn't matter because the decoder is never used directly anyway, and this is only a "feature learning" stage of sorts?

- Why use the L1 loss on spectrograms?

- The recent work on Parallel WaveNet may allow for speeding up WaveNet when used as a vocoder, this could be worth looking into seeing as inference speed is used as an argument to choose different vocoder strategies (with poorer audio quality as a result).

- The title heavily emphasizes that this model can do multi-speaker TTS with many (2000) speakers, but that seems to be only a minor aspect that is only discussed briefly in the paper. And it is also something that preceding systems were already capable of (although maybe it hasn't been tested with a dataset of this size before). It might make sense to rethink the title to emphasize some of the more relevant and novel aspects of this work.


----

Revision: the authors have adequately addressed quite a few instances where I feel motivations / explanations were lacking, so I'm happy to increase my rating from 6 to 7. I think the proposed title change would also be a good idea.

---

> ### Author Response · Authors · 2018-01-05
> **Thank you for the detailed comments and suggestions; they are really helpful to improve the quality of our paper.**
>
> To incorporate your and other reviewers' suggestions, we have expanded discussions on the motivations behind the design choices in this paper. For example:
> 1) We have added a new section ("Convolution Blocks for Sequential Processing") to motivate the architecture design choices for the convolution blocks used in our model.
> 2) In the "Encoder" section, we have added the motivation behind mixing key vector h_k and embedding h_e.
> 3) In the "Decoder" section, we have expanded the explanation of query generation for attention and explain the motivation to use L1 loss.
> 4) In the "Attention Block" section, we have added more explanations for our attention mechanism choices, attention's role in the overall architecture, the choice of positional encodings, and techniques to minimize attention errors.
> 5) In the "Converters" section, we have added clarification and justification for the relationship between the decoder hidden state and the converter/vocoder.
>
> We note that due to the required additions, our page limit has exceeded the suggested.
>
> Other comments:
>
> - "The separation of the "decoder" and "converter" stage is not entirely clear to me ..."
> * For a complex deep learning model like a TTS system, it can be challenging to train end-to-end in practice - instead, auxiliary/intermediate losses and multi-task learning may be preferred to guide the training of whole system. In decoder architecture, the loss for mel-scale spectrogram generation guides training of the attention mechanism, because the parameters are trained with the gradients from mel-scale spectrogram generation besides vocoder parameter generation. Our experiments suggest that a mel-scale spectrogram is a compact audio representation with sufficient information content to train a robust attention mechanism. Using mel-scale spectrograms yields fewer attention mistakes, compared to other high-dimensional audio representations (e.g., linear spectrogram, or other vocoder parameters). We observe that inputting the last hidden states of the decoder rather than mel-scale spectrograms to the converter network yields slightly higher audio quality. We attribute this to the richer information content of the hidden states, as a mel-scale spectrogram is a fixed representation.   Since WaveNet is conducive to producing high quality audio directly from mel-scale spectrograms, for WaveNet vocoder, we use mel-scale spectrograms as the external conditioners to the WaveNet architecture.
>
> - "At the bottom of page 2 it is said that "the whole model is trained end-to-end, excluding the vocoder" ... "
> * We have removed this phrase.
>
> - "In Section 3.3, the point of mixing of h_k and h_e is unclear to me. Why is this done?"
> * We have added an explanation for this design choice: "The attention value vectors are computed from attention key vectors and text embeddings, $h_v = \sqrt{0.5} (h_k + h_e)$, as in  (Gehring et al., 2017), to jointly consider the local information in $h_e$ and the learned long-term context information in $h_k$."
>
>  - "The gated linear unit in Figure 2a shows that speaker embedding information is only injected in the linear part. Has this been validated to work experimentally better ... "
> * We have compared various alternatives to make convolution blocks speaker-dependent, including adding speaker-dependent biases and/or gains. The particular choice in the paper has yielded the best results empirically.
>
> - "When the decoder is trained to do autoregressive prediction of spectrograms, is it autoregressive only in time, or also in frequency? ... "
> * The prediction of spectrogram is autoregressive only in time, so there is an implicit conditional independence assumption across frequency bins given all past timesteps. This design choice is important to achieve faster inference, and it yields good enough result as we demonstrated.  As you mentioned, converter network plays a more important role in determining the audio quality, and it is non-causal (and hence is not autoregressive).
>
> - “Why use the L1 loss on spectrograms?”
> * Prediction of spectrograms is treated as a regression problem. We choose L1 loss since it yields the best result empirically. Other common regression loss functions such as L2 loss may suffer from outlier spectral features (which may correspond to non-speech noise). We have clarified this point in Section 3.5.
>
> - "The recent work on Parallel WaveNet .... "
> * Thanks for pointing it out. Parallel WaveNet can be integrated as a vocoder, which may yield better audio quality while still achieving fast inference. We think it is an important future direction and leave it to future work.
>
> - "... It might make sense to rethink the title to emphasize some of the more relevant and novel aspects of this work."
> * Thanks for your suggestion. We are considering to change the title to "Deep Voice 3: Scaling Text-to-Speech with Convolutional Sequence Learning".

---

### Decision · Program_Chairs · 2018-01-29
**ICLR 2018 Conference Acceptance Decision**

**Decision:**

Accept (Poster)

**Comment:**

The paper describes a production-ready neural text-to-speech system. The algorithmic novelty is somewhat limited, as the fully-convolutional sequence model with attention is based on the previous work. The main contribution of the paper is the description of the complete system in full detail. I would encourage the authors to expand on the evaluation part of the paper, and add more ablation studies.